# Use of Isothermal and Isoperibolic Calorimetry to Study the Effect of Zinc on Hydration of Cement Blended with Fly Ash

**DOI:** 10.3390/ma13225215

**Published:** 2020-11-18

**Authors:** Pavel Šiler, Iva Kolářová, Radoslav Novotný, Jiří Másilko, Jan Bednárek, Martin Janča, Jan Koplík, Jan Hajzler, Lukáš Matějka, Michal Marko, Jiří Švec, Martin Zlámal, Eva Kuzielová, Tomáš Opravil, František Šoukal

**Affiliations:** 1Materials Research Centre, Faculty of Chemistry, Brno University of Technology, CZ-61200 Brno, Czech Republic; Kolarova.ivuska@seznam.cz (I.K.); xcnovotny2@fch.vut.cz (R.N.); masilko@fch.vut.cz (J.M.); xcjancam@fch.vut.cz (M.J.); koplik@fch.vut.cz (J.K.); xchajzlerj@fch.vut.cz (J.H.); xcmatejkal@fch.vut.cz (L.M.); xcmarkom@fch.vut.cz (M.M.); opravil@fch.vut.cz (T.O.); soukal@fch.vut.cz (F.Š.); 2Institute of Environmental Technology, Technical University of Ostrava, CZ-70800 Ostrava, Czech Republic; jan.bednarek@vsb.cz; 3Faculty of Civil Engineering, Institute of Concrete and Masonry Structures, Brno University of Technology, CZ-60200 Brno, Czech Republic; svec@fch.vut.cz (J.Š.); zlamal.m@fce.vutbr.cz (M.Z.); 4Institute of Construction and Architecture, Slovak Academy of Sciences, SK-845 03 Bratislava, Slovak; eva.kuzielova@savba.sk

**Keywords:** portland cement, zinc, isothermal calorimetry, isoperibolic calorimetry, fly ash

## Abstract

Increasing utilization of secondary raw materials and alternative fuels results in increasing contents of metals in cements. Zinc is one of these elements. It comes to cement with secondary raw materials such as slag or fly ash or by the utilization of used tires as an alternative fuel. Zinc ions significantly prolong the hydration process in cement. This work deals with the influence of zinc ions in the form of very poorly soluble ZnO salt and easily soluble ZnCl_2_ and Zn(NO_3_)_2_ on the hydration of cement blended with fly ash. Zinc was dosed in the range of 0.05%, 0.1%, 0.5% and 1% of cement weight. The effect of zinc on hydration was monitored by isothermal and isoperibolic calorimetry. A 15% addition of fly ash to cement mainly causes further retardation of hydration reactions due to the reactions of fly ash particles with Ca^2+^ ions from cement. The strongest effect on the hydration retardation from all investigated compounds showed in ZnO as it dissolves very slowly. On the contrary, for the dosage of 1% of zinc in the form of ZnCl_2_ significant acceleration of hydration occurred. In this work, a synergistic effect on the prolongation of hydration with a combination of cement, zinc and fly ash was demonstrated. The lengths of induction periods were assessed from detected calorimetric curves and from these lengths the curves were gained by fitting with the exponential function. Final products were next analyzed using X-ray diffraction.

## 1. Introduction

Portland cement is the most used type of cement for the production of concrete and mortars. It consists of mainly the mixture of oxides of calcium, silicon, aluminum and iron. Portland cement and similar materials are produced by burning limestone (the source of calcium) with clay or sand. The main phases formed during the production of cement are tricalcium silicate (C_3_S-3CaO·SiO_2_ or Ca_3_SiO_5_, alite), dicalcium silicate (C_2_S-2CaO·SiO_2_ or Ca_2_SiO_4_, belite), tricalcium aluminate (C_3_A-3CaO·Al_2_O_3_ or Ca_3_Al_2_O_6_, celite) and calcium aluminoferrite (C_4_AF-4CaO·Al_2_O_3_·Fe_2_O_3_ or Ca_4_Al_2_Fe_2_O_10_, brownmillerite) [1,2,3]. Gypsum (CaSO_4_·2H_2_O) is added to clinker as a setting agent, which reacts with C_3_A and Ca(OH)_2_ (CH, portlandite) to form ettringite (C_6_A$\bar{S}$_3_H_32_) [4].

The cement industry is one of the largest consumers of energy and natural resources. The endeavor is to substitute these resources with alternative materials. These alternatives mostly contain trace elements which can incorporate into the clinker phases during its production and further affect the cement properties [5]. Among many elements which can negatively affect cement properties, zinc is of great importance. Its main sources are alternative fuels, e.g., already mentioned tires and other rubber materials and also cement alternatives such as metallurgy slags or fly ash [6]. Zinc is known mainly for the retardation of cement hydration, for the change of structure and for strong decrease of initial strength [7,8].

The most common theory describing the mechanism of setting retardation by zinc ions suggests the formation of a layer of poorly soluble hydroxides on the surface of hydrated grains, which forms a physical barrier for water transport towards clinker minerals. According to recent knowledge the main reason for the setting retardation is the conversion of hydroxide to zincate. This reaction leads to the consumption of Ca^2+^ and OH^‒^ ions from the pore solution which consequently inhibits the formation of C-S-H gel [2,9,10] and thus the hydration of silicate phases up to total hydrolysis and the conversion of clinker is inhibited. The reactions leading to the formation of zincate are given in Equations (1)–(3) [5,11]:(1)$${\mathrm{Zn}}^{2+}+2{\mathrm{OH}}^{-}\to \mathrm{Zn}{\left(\mathrm{OH}\right)}_{2}$$
(2)$$\mathrm{Zn}{\left(\mathrm{OH}\right)}_{2}+2{\mathrm{OH}}^{-}\to 2H_{2}O+{\mathrm{ZnO}}_{2}^{2-}$$
(3)$$2{\mathrm{ZnO}}_{2}^{2-}+C_{3}S/O-{\mathrm{Ca}}^{2+}+6H_{2}O\to C_{3}S/O-{\mathrm{CaZn}}_{2}{\left(\mathrm{OH}\right)}_{6}\cdot 2H_{2}O+2{\mathrm{OH}}^{-}$$

Another introduced theory is the poison effect. According to this theory, the retardation by zinc, which is generally considered as a negative effect, is caused by the poison effect, by which the nucleation of hydration products is delayed or blocked. The poison improves the supersaturation of Ca(OH)_2_ and C-S-H in pore solution. After the poison effect is overcome, faster nucleation and growth occurs, compared to pure cement paste [8]. Odler and Schmidt [12] studied the influence of zinc on the formation of clinker minerals and the distribution of zinc in between the Portland cement phases. They discovered that the free lime content decreases with increasing temperature and burn-out time as well as with increasing dosage of zinc oxide. Zinc cations catalyze the reactions of clinker minerals formation. Increasing content of zinc changes, also, the phase distribution of clinker. The contents of alite (C_3_S) and interstitial phases (C_3_A and C_4_AF) increase at the expense of belite (C_2_S). The authors found two different reasons for this phenomenon. The first is the substitution of calcium ions by zinc ions and hence the release of more calcium ions for further reaction with silicates and the second one is increased content of C_4_AF at the expense of C_3_A.

The effect of zinc on the differences in the surface composition, surface morphology and osteogenesis performance of the calcium phosphate cement hydration product were studied at work [13].

Gineys et al. [5], in their work, discovered that zinc reacts preferentially with aluminates and affects the stability of C_3_A phase forming a new compound Ca_6_Zn_3_Al_2_O_12_. The limit of zinc content was described as the concentration at which the content of C_3_A phase is decreased and replaced by Ca_6_Zn_3_Al_2_O_12_ phase. The reactivity of C-Z-A phases was tested by Arceo H. and Glasser [14]. They prepared pure Ca_6_Zn_3_Al_2_O_12_ phase. Based on the results it can be assumed that despite the Ca_6_Zn_3_Al_2_O_12_ phase being reactive, zinc ions are blocked in insoluble hydrates.

A similar experiment was performed by Gineys et al. [6]. They found out that insoluble hydrates are formed on the surface of zinc phase and hence there are only limited reactions of zinc with C-S-H gel. One of given explanations is non-optimized gypsum content, which was based on the content and reactivity of C_3_A phase because overdosage in gypsum could induce a delay in cement hydration [15].

Li et al. [16] studied the effect of zinc chloride and sulfate on the hydration and final compressive strengths of high performance concrete (HPC). They found out that the setting retardation is stronger with chlorides than with sulfates.

Šiler et al. [17] dealt with the effect of zinc in the form of ZnO, Zn(NO_3_)_2_ and ZnCl_2_ on cement pastes and in [18] they doped cement with slag in the same experiments. They discovered that ZnO is the strongest retarder due to low solubility. In the cement paste the hydration was accelerated when 1% ZnCl_2_ was added, but when combined with slag, the acceleration did not occur. Next, the synergic effect was observed, which retarded the hydration when zinc was combined with slag.

Fly ash (FA) is the product of coal combustion in thermoelectric plants [19,20]. The term fly ash was accepted as it is transported from combustion chamber by exhaust fuels. Fly ash can be classified as cementitious or pozzolanic [21,22]. The cementitious fly ash sets when mixed with water. The pozzolanic fly ash hardens when activated by alkaline substance prior to mixing it with water. The alkaline substance can be e.g., quick lime, commonly present in cementitious mixtures. These features predetermine fly ash to be used as a cement replacement in concrete and many other building applications. Fly ash contains many heavy metals, especially zinc, lead and copper [23]. Primary fly ash can contain 0.8 wt.% Zn [24]. Secondary fly ash, which is formed by melting the primary fly ash, can contain 2.2% to 20.7% Zn [23]. Fly ash is one of possible zinc donors for cementitious materials.

This work shows the effect of zinc in the form of ZnO, Zn(NO_3_)_2_ and ZnCl_2_ in combination with fly ash on the hydration of Portland cement. This behavior is studied by means of isothermal and isoperibolic calorimetry. The observation of the behavior of various compounds of zinc combined with various secondary raw materials is important for further utilization of cement contaminated by zinc in various applications. The scientific importance of this work consists particularly in demonstration and quantification of effect of used components on the hydration of Portland cement, in identification of newly formed compounds by means of XRD and in the explanation of length of induction period by mathematical function. Obtained results can help optimize the amount of used secondary raw materials in cements containing zinc.

## 2. Materials and Methods

### 2.1. Materials and Sample Preparations

The mixtures were prepared from the cement CEM I 42.5 R Mokrá–Českomoravský cement, a.s., Heidelberg Cement, Czech Republic (x_10_ = 0.47 µm, x_50_ = 8.89 µm, x_90_ = 34.42 µm). Fly ash was from ČEZ Energetické produkty, s.r.o., Hostivice, Czech Republic (x_10_ = 8.36 µm, x_50_ = 112.57 µm, x_90_ = 298.37 µm). The particle size was determined by laser-diffraction method. The chemical and phase compositions of the components used are shown in Table 1 and Table 2.

Zinc was added in the form of poorly soluble ZnO and well soluble compounds Zn(NO_3_)_2_∙6H_2_O and ZnCl_2_. The amount added was in the range from 0.05 to 1 wt.% of the cement substitution (the percentages of the replacement were always calculated to pure zinc amount in binder). The pastes were mixed with distilled water with water to binder ratio equal to 0.4.

A 15 wt.% FA replacement of cement was chosen to study the effect of FA on the hydration of cement doped with zinc.

For the measurements in the isothermal calorimeter, the amount of 7 g of mixture was dosed into glass ampoule and placed in the device, where it was immediately mixed. For the measurements in isoperibolic calorimeter the amount of 300 g of pre-mixed mixture was placed in Styrofoam cup provided with a thermo-insulating jacket and a thermocouple and this mixture had been monitored until the main reaction of silicates was finished and the mixture temperature reached the ambient temperature.

Due to the sensitivity of measurement of X-ray diffraction analysis (XRD) the measurement was performed with samples containing 5 wt.% of zinc. Right after mixing, the samples were placed in Styrofoam cups and stored in a humid environment until the time of measurement. Next the samples were ground in a vibrating mill and the hydration was stopped by rinsing the sample with acetone. The samples were dried at the temperature of 50 °C to remove residual acetone.

### 2.2. Calorimetry

For the monitoring of zinc effect on the hydration the methods of isothermal and isoperibolic calorimetry were used. The main differences in results were caused mainly by the measurement conditions since in the case of isothermal calorimetry the measurement was performed under exactly defined conditions, while the isoperibolic measurements proceed under real conditions. The combination of those two methods brings important information about the course of particular reactions (isothermal calorimetry) and the comparison with the process in real environment (isoperibolic calorimetry) [25].

The prediction and control of concrete temperature rise due to cement hydration is important for mass concrete structures since large temperature gradients between the surface and the core of the structure can lead to cracking, thus reducing durability of the structure [26].

### 2.3. Isothernal Calorimetry

This method is often used for the monitoring of hydration of hydraulic binders, pozzolan materials and latent hydraulic properties [27,28]. The measurement proceeded at an exactly set temperature inside the device together with reference sample (in this case the quartz sand).

Due to exactly defined conditions this method can be used for long lasting measurements. By the changing of temperature, the reaction rate can be changed and, hence, particular reactions can be studied in more detail, especially during early stage of hydration where the heat rate is relatively high [17].

The measurement of hydration heat evolution was carried out using isothermal calorimeter TAM Air (TA Instruments, New Castle, DE, USA). The measurement was based on ASTM C 1679 [29]. Quartz sand was used as a reference. The measurement was carried out at 25 °C. The introduction of the sample preparation is mentioned below.

### 2.4. Isoperibolic Calorimetry

Isoperibolic calorimetry is a method which is often used for the monitoring of hydration of hydraulic binders, especially cements [30,31,32]. The principle is the measurement of temperature changes during hydration at constant ambient temperature. For the calculation of total released heat, the subsequent numerical data integration is performed. Data required for this calculation was gained from the device calibration [25]. The hydration heat is evolved during the hydration and the sample heats up. During isoperibolic calorimetry the same heating occurs during the hydration under real conditions. For the study of hydration reactions, it is, therefore, advantageous to use a combination of both these calorimetric methods [25].

### 2.5. X-ray Diffraction Analysis (XRD)

XRD is a non-destructive method used for testing the crystallic materials. X-ray diffractogram is a measurement of the characteristics of investigated substance. It provides the information which can be used for the determination of quality and quantity of various crystalline phases [33,34]. The X-ray diffraction analysis was performed with X-Ray diffractometer Empyrean (Panalytical) (Malvern Panalytical, Royston, UK) with Bragg–Brentano parafocusing geometry and using CuKa radiation. The measurements were done within the range from 5 to 120°2θ with angular step of 0.013°2θ and 25 s duration using automatic divergence slits to maintain the constant irradiation of the sample area. The measurements were repeated four times and then summed.

## 3. Results and Discussion

The addition of fly ash to cement mainly brings about further delay of hydration reactions due to the reaction of fly ash particles with Ca^2+^ ions from cement. While the reactivity at lower temperature in isothermal calorimeter is slower, also, a lower heat evolution occurs, which is due to the substitution of very reactive cement grains by less reactive particles of fly ash. Even the pozzolanic reaction in later phases of hydration did not produce such high values of heat evolved for any monitored sample, as to reach the values of samples containing solely cement. On the contrary, in the real environment of the isoperibolic calorimeter, where the sample heats up, higher values of evolved heat were measured compared to samples with only cement due to faster course of pozzolanic reaction. The amount of evolved heat depends mainly on the content of free lime in fly ash, but, on the other hand, it also depends on the amount of reactive amorphous phase [30]. Fly ash mostly contains rather non-reactive crystalline phase. The reaction of all these components is affected by the temperature of the reaction environment. Increasing concentration of zinc results in the increase of the first peak of the hydration curve [27]. This rise is caused, mostly, by the reactions of aluminate phases. This increase can be caused either by the effect of zinc, as a smaller amount of ettringite is formed compared to the hydration of ordinary Portland cement (OPC), or by high content of amorphous phase in fly ash (78%), which can react faster thanks to another reaction mechanism due to the presence of zinc e.g., by the pozzolanic reaction [21,28,30,31].

When the concentration of zinc in the mixture is low the temperature is similar to that of reference sample REF III (15% of pure cement substituted by fly ash). Increasing content of zinc causes the decrease of temperature and consequently the retardation of hydration reactions. Furthermore, with increasing amount of zinc the number of reactive sites suitable for the incorporation of zinc ions into the membrane of hydrating grains decreases [35] and, thus, zinc reacts with ions present in pore solution. The total heat evolved during the isoperibolic measurement decreased with increasing amount of zinc. None of the samples from this series reached higher or similar value of heat compared to the reference sample Ref III. This was caused by the effect of zinc in amorphous phase [17,18], as well as in crystallic compounds Ca(Zn_2_(OH)_6_)(H_2_O)_2_ and Zn_5_(OH)_8_Cl_2_H_2_O which were discovered by means of XRD after 24 h.

The development of heat flow during the isothermal measurement is different from that during the isoperibolic measurement. For lower concentrations of zinc (to 0.5%) the values of heat flow were higher than those for Ref III. On the contrary, for the concentrations of zinc above 1 wt.% they were lower, for Zn(NO_3_)_2_∙6H_2_O they even dropped below the value of Ref III. The decrease of heat flow can be explained by higher amount of Zn^2+^ in amorphous phase because the formation of a zinc membrane (amorphous phase) reduces the reaction rates of some reactions, thereby reducing the heat flux.

The same behavior in heat evolution under isothermal conditions was observed for the samples with Zn(NO_3_)_2_∙6H_2_O and ZnO; the evolved heat reaches higher values than that of Ref III up to the zinc concentration of 0.5 wt.%. The mixtures with zinc content of 1 wt.% showed lower heat values than Ref III. The sample with ZnCl_2_ with the dosage of 1% of Zn showed accelerated hydration reactions, which consequently brought higher heat measured for this sample. For the rest of samples, we can expect increased total evolved heat above the value of reference sample in later stages of hydration due to pozzolanic reaction, the same as for the isoperibolic measurement. Here it must be pointed that the differences among all measured samples are very small. This very small deviation can be caused by inaccuracies during the measurement or also by the effect of zinc in samples. For example, in mixtures with ZnCl_2_, due to significant exothermic reaction while dissolving, high consumption of water could occur (water could either evaporate or it could be consumed for the formation of new phases containing higher amount of water), which was then missing for the hydration reactions.

According to obtained results, both calorimetric methods showed the strongest effect on the retardation of hydration observed for ZnO; already for the dosage of zinc of 0.1 wt.%. Moreover, in samples with fly ash a higher amount of non-reacted ZnO was observed, even after 7 days from mixing. Except for the samples with ZnO (0.5 and 1 wt.%) longer induction periods were measured in isothermal measurements. The heats evolved during induction periods increased with increasing concentration of zinc except for the sample with 1% of ZnCl_2_. The increase of heat can be caused by higher amount of doped compounds, but also by the precipitation of crystallic compounds Ca(Zn_2_(OH)_6_)(H_2_O)_2_, Zn_5_(OH)_8_Cl_2_H_2_O and Ca_2_Al(OH)_6_Cl(H_2_O)_2_.

### 3.1. CEM I + FA + Zn(NO_3_)_2_∙6H_2_O

#### 3.1.1. Isoperibolic Calorimetry

The data for the zinc nitrate sample are shown in Figure 1. For the samples with fly ash, the trend described in literature [18] was confirmed and lower value was measured compared to Ref I (by more than 15 °C) due to diluting effect. For two lowest zinc additions (0.05 and 0.1 wt.%) the maximum reached temperature in the system was similar to that of the sample Ref III (the difference was only 2 °C). However, the temperature generally decreases with increasing concentration of zinc. For the addition of 1 wt.% the maximum temperature decreased by 40% compared to Ref III. Due to the inhibiting action of zinc, the Zn^2+^ ions bond onto the sites within the membrane of hydrating grains [5,6,11], where they cause the retardation of hydration, furthermore the maximum temperature is decreased and, thus, the hydration reactions are prolonged. This mixture also showed low exothermal peak between 20 and 30 h after mixing, just like the sample with Zn(NO_3_)_2_∙6 H_2_O with the same zinc concentration investigated in literatures [17,18]. From the X-ray diffractograms, however, the nitrate analogue of monosulfate described in samples with slag [19] was discovered after 90 days. However, just after 24 h the compound Ca(Zn_2_(OH)_6_)(H_2_O)_2_ was detected. Hence, this low peak could mean the precipitation time of this compound.

Moreover, the sample Ref III reached by 80 J∙g^−1^ lower value of total evolved heat compared to Ref I. From [30], the sample with fly ash should reach lower values of heat than Ref I due to its lower reactivity. Higher value is probably caused by high content of amorphous phase (78%) and also due to pozzolanic reaction. In the figure below the decrease of evolved heat in all samples containing zinc compared to Ref III can be observed. The difference in evolved heat among the zinc concentrations was almost 55 J∙g^−1^. The total evolved heat can be affected by zinc as well as by NO_3_^−^ ions, which either stay in the solution or incorporate into the amorphous phase. Obviously, the analogue of monosulfate 3CaO∙Al_2_O_3_∙0.83Ca(NO_3_)_2_∙0.17Ca(OH)_2_ can precipitate in a very small amount also in this case.

#### 3.1.2. Isothermal Calorimetry

The data for the zinc nitrate sample are shown in Figure 2. In samples with zinc concentration of 0.05 and 0.1 wt.%, a small sulfate depletion peak was recorded on the hydration curve together with the main hydration peak [27], same as in reference samples in literature [17,18]. As the zinc content rises, the first peak rises too and the second peak decreases up to the zinc concentration of 0.5 wt.%, where only one peak is present. This phenomenon can be explained by the inhibition of hydration reactions itself, and also by following acceleration of these reactions (narrowed peaks), probably due to the change of pH of the environment caused by the precipitation of Ca(Zn_2_(OH)_6_)(H_2_O)_2_.

The addition of fly ash causes slight decrease of heat flow by 0.7 mW∙g^−1^. Increasing zinc concentration up to 1 wt.% increases, also, the heat flow. After this value, significant drop by 1.2 mW∙g^−1^ below the value of Ref III was observed. In the samples with zinc concentrations of 0.05; 0.1 and 0.5 wt.% higher values of heat flow were measured, compared to Ref III. The decrease of heat flow in mixtures with higher content of zinc is probably caused by the incorporation of higher amounts of zinc into amorphous structures. Thus, the induction period is prolonged. For the addition of zinc of 1% another exothermic peak was recorded around 22 h, same as during isoperibolic calorimetry. This peak is probably induced by the precipitation of Ca(Zn_2_(OH)_6_)(H_2_O)_2_, as was already explained in previous works [17,18].

The sample containing fly ash released less heat by 20 J∙g^−1^ after 140 h than Ref I. Increasing concentration of zinc results in an increased amount of total evolved heat compared to Ref III. Only in the sample containing 1 wt.% of zinc the recorded heat was lower by 12 J∙g^−1^. At the same time all samples reached lower heat than Ref I. As the differences in evolved heats among samples containing zinc and Ref III up to 20 J∙g^−1^, can be caused by the inaccuracies during the measurement, it can be assumed that zinc does not significantly affect total evolved heat.

### 3.2. CEM I + ZnCl_2_

#### 3.2.1. Isoperibolic Calorimetry

The data for the zinc chloride sample are shown in Figure 3. In samples containing 1% of zinc in the form of ZnCl_2_ a noticeable effect on the acceleration of hydration caused by Cl^−^ ions was observed. The acceleration occurred under both, isothermal as well as real conditions, compared to pure cement [17], where this effect was observed only under real conditions during isoperibolic measurement—contrary to data introduced in literature [17], where no peak was found on the calorimetry curve for 1 wt.% of zinc in the mixture, probably due to a low temperature in a calorimeter (25 °C). This effect was observed in pure cement, but not in samples with slag [18]. The reason is probably higher amount of alkali ions which make the course of hydration reactions in fly ash easier. Another reason is increased permeability and diffusivity of chloride anions which is due to smaller ion radius than OH^−^ can cause the increase of pressure gradient and further deterioration of layer of C-S-H gel [36,37]. In the chart presenting the measurement by isoperibolic calorimetry, the rise of initial peak [27] with increasing zinc concentration can be observed. Only small temperature difference was measured between the sample Ref III and sample containing 1 wt.% of zinc. For the lowest (0.05 and 0.1 wt.%) zinc concentrations the measured temperature was comparable to that of Ref III. Further increase of zinc brings about stronger decrease of temperature. The drop in temperature maxima is due to the inhibition of hydration reactions, the membrane around the cement grains is modified to a larger extent. Due to lower temperature, the following hydration reactions are slower (the peak is lower and wider). The mixtures with zinc except for that with 1% of Zn reach lower values of heat than Ref III but higher than Ref I. The difference in released heat values between the samples with the lowest (0.05 wt.%) and the highest (1 wt.%) zinc content was 46 J∙g^−1^. The heat decrease can be caused by affecting the pozzolana reaction or by the incorporation of Zn^2+^ to amorphous structures of composite paste. Additionally the influence of Cl^−^ ions cannot be omitted, as despite being incorporated into crystallic compounds Zn_5_(OH)_8_Cl_2_H_2_O and Ca_2_Al(OH)_6_Cl(H_2_O)_2_, proved by XRD, they partly stay either in pore solution or incorporate into amorphous structures as well.

A significantly prolonged induction period compared to Ref III was observed already for the sample with 0.05 wt.% of zinc, particularly after 5.7 h. Due to added ZnCl_2_, the heat evolved during the induction period rises. The sample with 0.5 wt.% of zinc produced by almost 44 J∙g^−1^ higher heat than Ref III. The sample with the highest zinc concentration showed slight decrease of heat evolved during the induction period, compared to the mixture containing 0.5 wt.%, which agrees with the results with slag in literature [19]. Gradual increase of heat during the induction period is due to increasing content of ZnCl_2_, which shows exothermal dissolving in cement paste. Moreover, the precipitation of Zn_5_(OH)_8_Cl_2_H_2_O and Ca_2_Al(OH)_6_Cl(H_2_O)_2_ can partly influence the heat evolution. These compounds were detected by XRD after just 24 h.

Three consistent trends were observed for the mixtures containing Zn(NO_3_)_2_∙6H_2_O and ZnCl_2_. The first one is the prolongation of induction period with increasing content of zinc. Secondly, the decrease in maximum temperatures was detected for the samples containing 0.5 and 1 wt.% of zinc. The third visible trend was the release of smaller amount of total heat compared to Ref III. The value of heat can be affected by the incorporation of both, pure zinc ions as well as zinc compounds anions into the amorphous phase of fly ash or, also, the pozzolana reaction can be affected. Except for the samples containing 0.5 wt.% of zinc, the retardation times for all samples from both sets are approximately 2 h. That means they had very similar inhibitory behavior of soluble compounds. Except for the sample containing 1 wt.% of zinc in the form of ZnCl_2_, higher evolved heats were observed during the induction period in samples containing ZnCl_2_. That corresponds to higher heat evolved during the dissolution of ZnCl_2_ as well as to the formation of Zn_5_(OH)_8_Cl_2_H_2_O and Ca_2_Al(OH)_6_Cl(H_2_O)_2_ compounds. In mixtures containing Zn(NO_3_)_2_∙6H_2_O, only Ca(Zn_2_(OH)_6_)(H_2_O)_2_ in small amounts was detected after 24 h.

Comparing the values of maximum temperatures and times when they were reached in samples with soluble compounds it can be read from charts one and two, that the values of maximum reached temperatures are comparable. For Zn(NO_3_)_2_, accelerated course of hydration can be observed, except for the sample with 1% which exhibited significant acceleration due to chloride ions. The phenomenon of accelerated hydration in the presence of Zn(NO_3_)_2_ is probably caused by the acceleration of setting due to catalysis of belite hydration by nitrate ions [38,39].

The values of total released heat are similar for soluble compounds. The exception is the concentration of zinc of 1% due to the acceleration by Cl^−^ ions.

#### 3.2.2. Isothermal Calorimetry

The data for the zinc chloride sample are shown in Figure 4. For all observed zinc concentrations, the heat flow is increased compared to Ref III, but none achieved the value of Ref I. Increased content of zinc increases the heat flow maximum. The differences in heat flows for the samples containing zinc in the form of ZnCl_2_ are very small. Total released heat values in the samples containing 0.05 and 0.1 wt.% of zinc after 140 h were comparable to Ref III, the values were lower by 4 J∙g^−1^. Increasing amount of zinc decreased the released heat. The difference in heats between samples containing 0.05 and 1 wt.% of zinc makes only 20 J∙g^−1^. These very small differences could be caused by inaccuracies during the measurements.

The values of total released heat are very similar for all measured samples. It means that the value of total released heat in samples containing ZnCl_2_ is not really influenced by its content. Due to increasing the amount of zinc incorporated into amorphous structures which affects normal course of hydration, the inhibitory effect rises. The mixtures with 0.05 and 0.1 wt.% of zinc released similar amount of heat during the induction period as the reference sample with fly ash. Increasing concentration of zinc causes increasing released heat except for the sample with the highest concentration, in which the recorded heat value was lower than that of a sample containing 0.5 wt.% of zinc. Similar values in mixtures with small content of zinc and in reference sample are caused, as already mentioned, by the sample preparation, when a given amount of ZnCl_2_ was dissolved in mixing water and next it was dosed into the mixtures. On the contrary, low heat values in mixtures with high dosages of ZnCl_2_ can be explained by insufficient homogenizing of sample and its less fluid consistency.

When the heat flows of soluble compounds are compared, it can be seen that the curves of samples with concentrations of zinc up to 0.5% in the form of soluble compounds are very similar. The difference is observed for the curve belonging to the sample with 1% of Zn, as ZnCl_2_ exhibits the acceleration of hydration due to Cl^−^ ions. This acceleration, however, occurs only at higher ZnCl_2_ concentration. Due to this acceleration the sample with 1% of ZnCl_2_ reaches the main hydration peak before the sample with 0.5% of ZnCl_2_, contrary to Zn(NO_3_)_2_ in which, due to high inhibitory effect in the 1 wt.% zinc sample, lower maximum temperature development and, therefore, slower hydration reactions occur (lower and wider peak). In samples with ZnCl_2_ lower values of heat flow maxima were observed, which is consistent with the results given in literature for pure cement [17] and for cement with slag [18].

The values of total released heat are similar for both soluble compounds; however, for ZnCl_2_ the differences for various concentrations are very small.

### 3.3. CEM I + ZnO

#### 3.3.1. Isoperibolic Calorimetry

The data for the zinc oxide sample are shown in Figure 5. The same as in pure cement [17] and in combination with slag [18] the strongest hydration retardation occurred in mixtures with ZnO, especially due to slow dissolution of this compound. The samples containing 0.05, 0.1 and 0.5 wt.% of zinc reached similar values of maximum temperature like Ref III. The mixture with 1% of ZnO reached the lowest temperature. The content of zinc up to 0.5% did not significantly affect the temperature maxima. Higher content of zinc means more Zn^2+^ ions in amorphous phase and thus stronger inhibition effect.

For all tested concentrations lower heat was recorded than for Ref III, but higher than for Ref I. The samples with zinc content of 0.05 and 0.1 wt.% reached similar values of released heat after 160 h of hydration. Higher dosages of zinc gradually decrease the heat. That can be caused by gradual dissolution of ZnO which was detected in the sample by XRD, even after 7 days. The highest zinc concentration postponed the start of setting by more than 110 h compared to Ref III.

Same as in the case of samples with Zn(NO_3_)_2_∙6H_2_O and ZnCl_2_ with zinc content of 0.05 and 0.1 wt.% the maximum temperatures are comparable to Ref III. Moreover, the mixtures with ZnO show noticeable decrease of temperature up to the zinc content of 1 wt.%. That might be caused by the absence of NO_3_^−^ and Cl^−^ anions as well as by the nature of ZnO compound.

The samples with 1 wt.% of zinc achieved for all observed compounds similar values of temperature, particularly about 33 °C with the deviation of 3 °C. Total released heat depends on the amount of zinc in samples. With increasing zinc content, it decreases in all tested compounds. The released heats in all observed mixtures with fly ash were lower than that of Ref III. This phenomenon can also be explained by the precipitation of new compounds containing either Zn^2+^ ions Ca(Zn_2_(OH)_6_)(H_2_O)_2_ and Zn_5_(OH)_8_Cl_2_H_2_O or chloride anion–Ca_2_Al(OH)_6_Cl(H_2_O)_2_.

ZnO up to the content of 0.1 wt.% affects the induction period the most significantly. The longest induction period of more than 110 h was recorded for the sample containing ZnO with 1 wt.% of zinc. This phenomenon could be explained by gradual dissolution of ZnO in alkaline environment providing longer time for the attack of zinc on the membrane which covers the cement grains. However, the precipitation of Ca(Zn_2_(OH)_6_)(H_2_O)_2_ already occurs within 24 h. Its amount rises, even up to 7 days, when all the doped amount of ZnO has reacted. Furthermore, the heat released during the induction period was the highest in the samples with ZnO, up to the zinc content of 0.1 wt.%. This could be assigned to the longest induction periods in these samples.

#### 3.3.2. Isothermal Calorimetry

The data for the zinc oxide sample are shown in Figure 6. Same as for the soluble salts of zinc also here the sulfate depletion peak occurs. For the dosage of ZnO the peak is visible only in the samples with 0.05 wt.% of zinc, at higher contents the peak disappears. Again, the main hydration peak decreases and the sulfate depletion peak increases. This phenomenon probably appears due to the effect of zinc on the induction period length. Its prolongation occurs due to the incorporation of zinc into amorphous structures, probably also the pH is influenced. Hence, after some time the effect of zinc is suppressed, probably due to the precipitation of Ca(Zn_2_(OH)_6_)(H_2_O)_2_ and faster course of hydration reaction occurs, which could be proved by the disappearance of sulfate peak as well as by narrowed peaks. Increasing amount of zinc up to the concentration of 0.1 wt.% causes the increase of achieved maximum heat flow. For the concentration of 0.5% the decrease can be observed due to slow dissolution of zinc, but the value is still higher than that for the reference sample. For the sample containing 1% of Zn the value drops below the reference value. Following, increase of heat flow could correspond to gradual reaction of total ZnO content.

Total released heat was recorded for the time of 170 h. None of the samples reached the values as high as that of Ref I after such long period of time. The mixtures containing 0.05 and 0.1 wt.% of zinc released the heat comparable to that of Ref III (higher by 3 J∙g^−1^). Zinc in the amount of 1 wt.% released significantly less heat than Ref III. We can suppose though, that during further hours the hydration would release further heat and that the final heats would reach similar values.

When the hydration courses measured by various methods are compared, it can be observed that due to slow dissolution of ZnO the course of hydration is the same for both, isothermal and isoperibolic calorimetry.

For all three tested compounds the sulfate depletion peak can be observed in samples containing 0.05 wt.% of zinc. This peak also occurs in mixtures containing 0.1 wt.% of zinc, but only in the form of soluble salts. Which means the strongest effect of zinc in the form of ZnO, which already in the content of 0.1 wt.% affects the hydration environment to such an extent that the following reactions proceed much faster. In mixtures with Zn(NO_3_)_2_∙6H_2_O and ZnO the sulfate depletion peak was higher and the main hydration peak was lower. The addition of ZnCl_2_ does not result in the rise of sulfate depletion peak above the main hydration peak. Moreover, due to the addition of Cl^−^ ion, higher concentrations the hydration reactions are accelerated. The maximum heat flows for the samples with concentrations up to 0.5 wt.% were higher than that of Ref III for all used compounds. Further increase of zinc content causes the maximum heat flow decrease. For Zn(NO_3_)_2_∙6H_2_O the maximum heat flow value even dropped below the value of Ref III. The increase of heat flow is probably caused by the incorporation of zinc into the newly formed structures of cement composite. However, when present in larger amount, free Zn^2+^ ions are present in the solution, which react with other ions from the pore solution and form the products which consume heat. The differences in released heat between all zinc compounds and Ref III do not exceed 25 J∙g^−1^, which is a very small deviation possibly caused by the inaccuracies during particular measurements. So, the addition of zinc only slightly affects the amount of total released heat.

When the calorimetric measurements of all three samples are compared, it can be stated, that for samples with ZnO, high extent of retardation can be observed, which was caused mainly due to slow dissolution of this compound. The dissolution also affects the amount of released heat, as in samples with higher concentration it starts rising steeply after 100 h of hydration. Additionally, it can be seen that in real conditions of isoperibolic measurement, almost all samples released higher heat than the sample of pure cement, except for the sample with 1% of ZnCl_2_ where achieved values were lower than that of the reference sample with fly ash. On the contrary, as for isothermal measurements the highest values of released heat were recorded for pure cement, while the samples containing fly ash reached similar values. The reason is probably a slow course of pozzolana reaction at lower temperatures.

### 3.4. The Comparison of Induction Period Length

The lengths of induction periods were determined from calorimetric measurements in the same way as in literature [17,18]. For the fitting of all curves the following equation was used:(4)$$y=A_{1}\cdot \exp\;\left(\frac{-x}{t_{1}}\right)+y_{0}$$
since the t_1_ value was negative for all samples, we can rewrite this as Equation (5).
(5)$$y=A_{1}\cdot \exp\left(\frac{x}{t_{1}}\right)+y_{0}$$

The resulting equations in Figure 7 are usually created from four points. The results from measurement for the sample containing 1% of Zn in the form of ZnCl_2_ was not included into the evaluation, since the acceleration of hydration, disallowing this way of evaluation, occurred in large extent. All curves created by fitting the recorded data with exponential function in Figure 7 exhibit only minimal deviations from real data. Statistical significance was checked for these equations.

From these curves, mainly similar behavior of soluble compounds, positive effect of Cl^−^ ions on the hydration acceleration and unambiguously the strongest retardation ability of ZnO due to its low solubility can be read. Comparing these curves to the results from literature [17] for slag, these curves are quite similar, for the combination with fly ash, however, the acceleration effect of Cl^−^ ions becomes more evident.

### 3.5. XRD Results

The following chapter shows the results by XRD analysis. This analysis provides the information about the phase composition of crystalline materials. The evaluation was performed by Rietveld Analysis of XRD Patterns. The results show the averages of four measurements. It must be taken into account that this analysis is not completely accurate and errors can reach up to 5%. Inaccuracies can also occur because the amorphous content present in cements is not taken into account. The results are shown in Table 3, Table 4, Table 5 and Table 6.

When compared to literature [18] it was discovered that in the sample with only fly ash and no Zn content similar amounts of main cement phases were present, but there was a slightly higher amount of ettringite and higher amount of calcite after 28 days. In Zn(NO_3_)_2_, a slightly lower amount of brownmillerite and an almost twice as high amount of ettringite were found. In the samples with ZnCl_2_, a higher amount of ettringite were found. For ZnO slightly higher amount of alite and brownmillerite, slightly higher amount of ettringite and lower content of portlandite detectable after 90 days just like for slag were found [18].

In samples without zinc ions, the development of hydration processes can be seen from 1st to 90th day of hydration, when there is a decreasing amount of clinker minerals (mainly alite) and an increasing amount of hydration products, especially portlandite. An increasing amount of calcite, which is formed by carbonation of portlandite, can also be seen. In contrast, for samples containing zinc ions, the hydration process is significantly slowed down. This is evident mainly from the content of the C_3_S phase, the content of which decreases only very slowly, and a more significant decrease often occurs only after 90 days of hydration. This would correspond to the absence of portlandite as one of the possible products of cement hydration. The lower amount of portlandite is caused by the formation of a zinc membrane, which changes the hydration mechanism.

Zinc was detected in the samples in small amount in the form of Ca(Zn_2_(OH)_6_)(H2O)_2_ except the sample with ZnCl_2_, which could be due to faster reactions in the presence of Cl^−^ ions. Furthermore, 3CaO. (Mg_0.67_Al_0.33_ (OH)_2_)(CO_3_)_0.165_(H2O)_0.48_ (hydrotalcite) is identified here, which is formed during fly ash reactions. Other minerals were identified only in individual samples.

## 4. Conclusions

The aim of this work was to study the effect of zinc in the form of soluble salts Zn(NO_3_)_2_, ZnCl_2_ and very poorly soluble ZnO on the hydration of Portland cement blended with 15% of fly ash.

A 15% addition of fly ash to cement mainly causes further retardation of hydration reactions due to the reactions of fly ash particles with Ca^2+^ ions from cement. Due to lower reactivity at lower temperature in isothermal calorimetry, lower heat evolution occurs which is caused by the substitution of very reactive cement grains by less reactive fly ash particles. Even the pozzolana reaction in later phase of hydration did not produce enough heat in any sample, to reach the same total released heat values as for the sample with pure cement. On the contrary, in real conditions of isoperibolic calorimeter, where the sample heats up, higher amounts of released heats were detected compared to pure cement, due to faster course of pozzolana reaction.

It has been shown that higher amounts of hydration-accelerating ions as Cl^−^ accelerate hydration. This effect is mainly presented by the amount of 1% ZnCl_2_ in which the hydration maximum of the second peak on the calorimetric curve was observed earlier than in the sample with 0.5% ZnCl_2_. These data were confirmed by both isothermal and isoperibolic calorimetry.

Furthermore, it has been found that when zinc is in the form of soluble compounds, the hydration proceeds faster than when it is in the form of insoluble compounds when gradual dissolution occurs associated with significant hydration retardation.

The lengths of induction periods were assessed from detected calorimetric curves and from these lengths the curves were gained by fitting with the exponential function. These curves comply well with measured data. Again, significant prolongation when insoluble ZnO is added and the acceleration of hydration for the addition of ZnCl_2_ can be observed.

From the curves detected by isothermal calorimetry it can be seen, that Zn(NO_3_)_2_∙6H2O and ZnO also increased sulfate depletion peak and decreased main hydration peak occur. The addition of ZnCl_2_ does not increase the sulfate depletion peak above the main hydration peak.

The effect of zinc on mechanical properties is not studied in this study. According to the available literature, it can be assumed that zinc reduces strength in the first days of hydration due to slowing hydration, but long-term strength may increase due to a different hydration mechanism and also, the possibility of zinc compounds acting as microfiller [40]. In the case of a combination with fly ash, long-term strengths may also increase due to a chemical (e.g., pozzolanic reaction) and physical effect (e.g., microfiller) [30].

## Figures and Tables

**Figure 1 materials-13-05215-f001:**
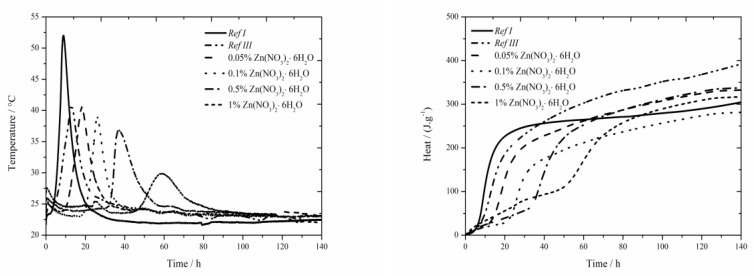
Isoperibolic for Zn(NO_3_)_2_ samples.

**Figure 2 materials-13-05215-f002:**
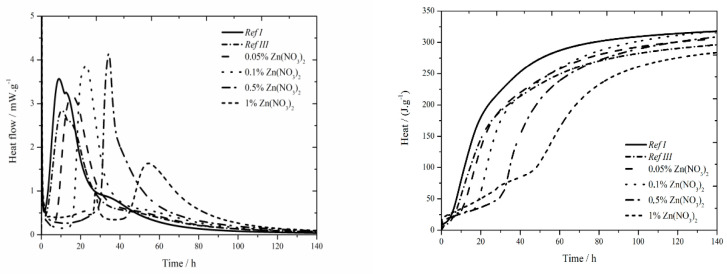
Isothermal curves for Zn(NO_3_)_2_ samples.

**Figure 3 materials-13-05215-f003:**
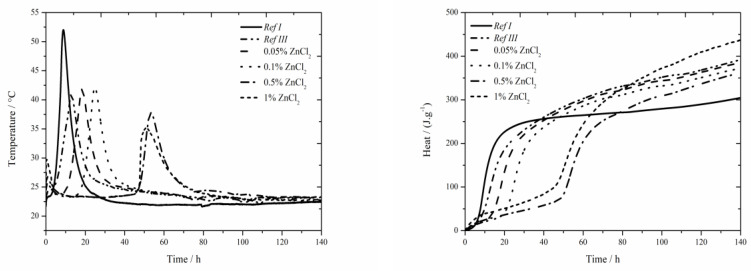
Isoperibolic curves for ZnCl_2_ samples.

**Figure 4 materials-13-05215-f004:**
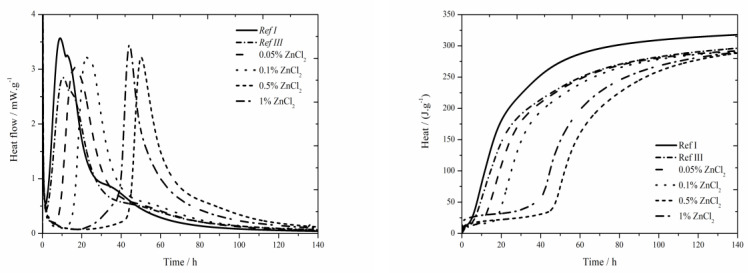
Isothermal curves for ZnCl_2_ samples.

**Figure 5 materials-13-05215-f005:**
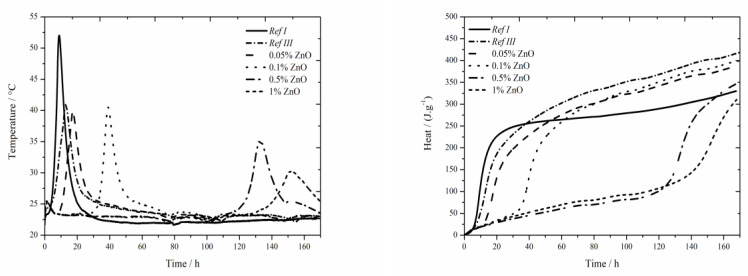
Isoperibolic curves for ZnO samples.

**Figure 6 materials-13-05215-f006:**
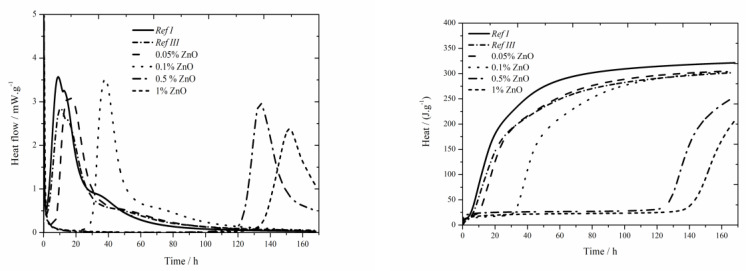
Isothermal curves for ZnO samples.

**Figure 7 materials-13-05215-f007:**
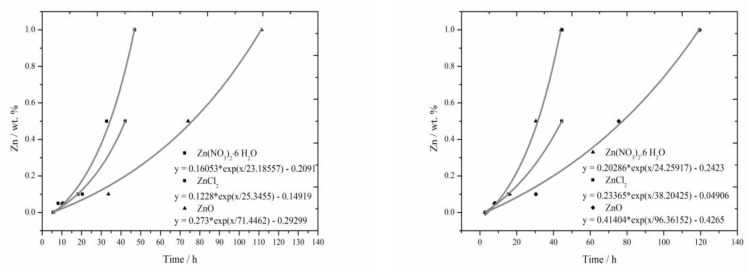
Graphical comparison of the lengths of the induction periods for isoperibolic (left) and isothermal (right) calorimetry.

**Table 1 materials-13-05215-t001:** Elemental composition of materials used.

Component	Cement%	FA%
CaO	65	9.70
SiO_2_	19	41.88
Al_2_O_3_	4	18.88
Fe_2_O_3_	3	24.34
MgO	1	1.8
SO_3_	3	-
S^2−^	0.04	-
Cl^−^	0.051	-
K_2_O	0.75	1.04
Na_2_O	0.15	0.41
ZnO	0.00005	-
LOI	3.1	1.10

**Table 2 materials-13-05215-t002:** Crystalline composition compositions of materials used.

Component	Cement%	FA%
C_3_S	67	N/A
C_2_S	11	N/A
C_3_A	7	N/A
C_4_A	11	N/A
Amorphous phase	N/A	78
Hematite	N/A	2
Mullite	N/A	27
Magnetite	N/A	3
Quartz	N/A	12
Crystobalite	N/A	5
Albite	N/A	5

**Table 3 materials-13-05215-t003:** Mineral composition of reference sample with cement and fly ash (FA).

Hydration (day)	*Ca_3_SiO_5_* Hatrurite (alite)	*Ca_2_(SiO_4_)* Larnite (belite)	*Ca_2_(Fe_2_O_5_)* Brownmillerite	*C_6_*$A\bar{S}$*_3_H_32_* Ettringite	*Ca(OH)_2_* Portlandite	*Ca(CO_3_)* Calcite	*(Mg_0.67_Al_0.33_(OH)_2_)(CO_3_)_0.165_ (H_2_O)_0.48_* Hydrotalcite	*SiO_2_*	*Fe_2_O_3_*
1	46.4	9.1	6.2	7.1	22.8	7.0	-	2.0	1.1
7	24.2	9.3	4.3	8.1	38.8	13.2	2.4	1.4	-
28	24.1	9.2	5.2	8.8	39.2	18.1	2.5	1.2	-
90	20.3	7.3	3.4	9.3	38.4	18.0	4.3	1.0	-

**Table 4 materials-13-05215-t004:** Mineral composition of sample with Zn(NO_3_)_2_∙6H_2_O.

Hydration (day)	*Ca_3_SiO_5_* Hatrurite (alite)	*Ca_2_(SiO_4_)* Larnite (belite)	*Ca_2_(Fe_2_O_5_)* Brownmillerite	*C_6_*$A\bar{S}$*_3_H_32_* Ettringite	*Ca(Zn_2_(OH)_6_)(H_2_O)_2_*	*Ca(CO_3_)* Calcite	*(Mg_0.67_Al_0.33_(OH)_2_)(CO_3_)_0.165_ (H_2_O)_0.48_* Hydrota.cite	*3CaO* *∙* *Al_2_O_3_* *∙* *0.83Ca(NO_3_)_2_* *∙* *0.17Ca(OH)_2_* *∙* *9.5H_2_O*	*SiO_2_*
1	53.7	12.3	3.1	18.0	1.2	9.2	2.2	-	1.1
7	52.8	12.2	2.9	18.2	2.3	10.3	3.6	-	1.1
28	52.9	11.8	2.8	19.1	2.4	10.5	2.4	-	1.3
90	47.9	12.0	2.8	18.6	2.8	10.1	-	8.9	1.2

**Table 5 materials-13-05215-t005:** Mineral composition of sample with ZnCl_2_.

Hydration [day]	*Ca_3_SiO_5_* Hatrurite (alite)	*Ca_2_(SiO_4_)* Larnite (belite)	*FeAlO_3_(CaO)_2_* Brownmillerite	*C_6_A*$\bar{S}$*_3_H_32_* Ettringite	*Ca_2_Al(OH)_6_Cl(H_2_O)_2_* Hydrocalumite	*Zn_5_(OH)_8_Cl_2_H_2_O* Siminkolleite	*SiO_2_*
1	58.5	11.3	3.3	13.2	3.6	2.0	1.0
7	53.9	11.4	3.2	14.6	4.2	3.3	1.2
28	53.8	12.1	3.3	14.5	5.4	2.3	1.1
90	53.3	11.2	3.4	13.0	7.1	2.2	1.9

**Table 6 materials-13-05215-t006:** Mineral composition of sample with ZnO.

Hydration(day)	*Ca_3_SiO_5_* Hatrurite (alite)	*Ca_2_(SiO_4_)* Larnite (belite)	*FeAlO_3_(CaO)_2_* Brownmillerite	*C_6_*$A\bar{S}$*_3_H_32_* Ettringite	*Ca(OH)_2_* Portlandite	*Ca(Zn_2_(OH)_6_)(H_2_O)_2_*	*ZnO*	*CaCO_3_* Calcite	*CaSO_4_∙2H_2_O* Gypsum	*SiO_2_*	*Ca(SO_4_)(H_2_O)_0.5_* Bassanite	*Ca_4_Fe_2_O_6_CO_3_·12 H_2_O*	*(Mg_0.67_Al_0.33_(OH)_2_)(CO_3_)_0.165_ (H_2_O)_0.48_* Hydrotalcite
1	59.2	15.3	4.0	6.3	-	2.2	6.3	9.2	3.1	1.1	-	-	-
7	54.0	16.1	4.9	9.4	-	4.2	6.2	8.3	-	1.1	2.1	-	-
28	47.4	13.2	3.2	10.3	-	5.0	-	8.4	-	2.9	-	3.2	-
90	46.9	13.1	3.1	9.5	9.2	3.9	-	9.5	-	2.0	-	-	3.6

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
