# Peer review of "Use of Isothermal and Isoperibolic Calorimetry to Study the Effect of Zinc on Hydration of Cement Blended with Fly Ash"

_materials, 2020, doi:10.3390/ma13225215_

Round 1

Reviewer 1 Report

The manuscript reports on the influence of Zn on the hydration of cement blended with fly ash. Though the rationale for the work is appreciable, it lacks in-depth to be considered for publication. Some additional studies must be supplemented.

Q1. The final products were just analyzed by XRD, which is surely not sufficient enough to arrive at a clear picture of the structural aspects of these cementitious materials. Some additional studies must be supplemented for the final product, such as Raman or NMR, or at least FTIR with a detailed analysis.

Q2. Some key pertinent references related to C-S-H materials were missed to be cited, such as J. Phys. Chem. C 2017, 121, 32, 17188–17196, J. Mater. Chem. A, 2018,6, 363-373, which needs to be added in the manuscript.

Author Response

Reviewer 1

The manuscript reports on the influence of Zn on the hydration of cement blended with fly ash. Though the rationale for the work is appreciable, it lacks in-depth to be considered for publication. Some additional studies must be supplemented.

Q1. The final products were just analyzed by XRD, which is surely not sufficient enough to arrive at a clear picture of the structural aspects of these cementitious materials. Some additional studies must be supplemented for the final product, such as Raman or NMR, or at least FTIR with a detailed analysis.

We tried to do an FTIR analysis. The results are summarized in the following link, but these results are not very convincing, so they were not added to the manuscript. If the reviewer wants to add these results to the manuscript, these results can be added

Due to the different prolongation of induction periods depending on the doped compound
containing zinc, the samples were left for analysis for 7 days in maturation baths. Furthermore, differences in the measured spectra with respect to the zinc concentration were monitored. For all samples bands typical of a hydrated cement sample were found. After 7 days the belt typical for O-H vibration from portlandite was detected. It means his presence in the samples, but only in very small amounts as it was not detected by XRD analysis. Also belts typical for water ν1, ν2 and ν3–O-H were also discovered. For mixtures with s Zn(NO)3∙6 H2O, a shift in vibration ν 2–O-H was found, probably due to the presence of crystalline water in the compound. The calcite measured in the XRD spectra was also found in the FTIR spectra. With increasing concentration of doped zinc, the peak corresponding to the asymmetric valence vibration (ν 3) Si-O was shifted. This shift could involve the incorporation of Zn2+ ions into the CSH gel. Another common change in the spectra was the vibration shift (ν3) SO42-. The most significant shift was recorded in samples with Zn(NO3)2.6H2O probably due to the incorporation of the anion NO3- into the monosulfate. Independently on the test compound, zinc was found in mixtures with a zinc concentration of 1 wt. % as band at 3,601 cm-1, which could correspond to the zinc hydroxide compounds present. This discovery would correspond to the theoretical assumptions of the formation of a modified membrane around the hydrating grains. This membrane disintegrates over time (it slows down hydration) and zinc is incorporated into both the crystalline phases (described in the X-ray diffraction chapter) and amorphous compounds (probably CSH).

Q2. Some key pertinent references related to C-S-H materials were missed to be cited, such as J. Phys. Chem. C 2017, 121, 32, 17188–17196, J. Mater. Chem. A, 2018,6, 363-373, which needs to be added in the manuscript.

Thank you, these references have been added to manuscript.

Reviewer 2 Report

 General comment

The topic is interesting and frequently investigated by many scientists. Therefore, it may be surprising that the authors decided to quote 8 of their works and did not notice other, new and important works. I propose to supplement the literature. Here are just a few suggestions:

  • Materials (2019), 12(12), 1942, https://doi.org/10.3390/ma12121942 (the article generally involves fly ash, its various sources, etc. )
  • Materials Science and Engineering: C (2019), Vol. 105, 110065, https://doi.org/10.1016/j.msec.2019.110065 (the article involves the subject of the assessed work)
  • Cement and Concrete Research (2015), vol. 72, pp. 128-136, https://doi.org/10.1016/j.cemconres.2015.02.023

Detailed comments

  • The abstract is not very interesting. I propose to supplement it with the most important accomplishments of this work.
  • The subsections 2.3 and 2.4 should be changed. In these chapters, the description involving the application of the methods is unnecessary (it is rather for popular science articles). I propose to describe the measurement method of the samples, etc.
  • Similarly, the subsection 2.5 requires a precise description of the test bench and the measurement of the number of samples, etc.
  • Generally speaking, the reader does not know what samples and how many samples were tested. It is also unknown what the result is (the average? From how many samples? What is the spread ?)
  • Please provide the full uncertainty budget: A type, B type, standard and extended uncertainty.
  • The equations 4 and 5 are presented in Figures 7. The verification of the statistical significance of the equations parameters is missing. Without this, it is not known whether the obtained parameters are the result of a coincidence or not.
  • The equations in Figure 7 were obtained for 3 or 4 points. Is it sufficient to draw conclusions?
  • Can the equations in Figure 7 be generalized in some way for future research (for research by other researchers) or are they just equations fitted to these historical results. If so, what for?

Author Response

Reviewer 2

The topic is interesting and frequently investigated by many scientists. Therefore, it may be surprising that the authors decided to quote 8 of their works and did not notice other, new and important works. I propose to supplement the literature. Here are just a few suggestions:

  • Materials (2019), 12(12), 1942, https://doi.org/10.3390/ma12121942 (the article generally involves fly ash, its various sources, etc. )

This citation has been added to manuscript

  • Materials Science and Engineering: C (2019), Vol. 105, 110065, https://doi.org/10.1016/j.msec.2019.110065 (the article involves the subject of the assessed work)

The sentence “The effect of zinc on the differences in the surface composition, surface morphology and osteogenesis performance of the calcium phosphate cement hydration product were studied at work [14].” was added to manuscript.

  • Cement and Concrete Research (2015), vol. 72, pp. 128-136,

This citation has been added to manuscript 

Detailed comments

  • The abstract is not very interesting. I propose to supplement it with the most important accomplishments of this work.

The abstract was supplemented according to the opponent's comments.

  • The subsections 2.3 and 2.4 should be changed. In these chapters, the description involving the application of the methods is unnecessary (it is rather for popular science articles). I propose to describe the measurement method of the samples, etc.

Description of sample preparation and measurement is in chapter 2.1. Materials and sample preparations. Chapters 2.3 and 2.4 provide a brief introduction to the calorimetry methods used and a brief explanation of the differences between these methods. I consider it important to explain to audience the reason for using these methods and the differences between them. Calorimetric data are the main content of this article.

  • Similarly, the subsection 2.5 requires a precise description of the test bench and the measurement of the number of samples, etc.

The number of measurements and important parameters are given in the mentioned chapter.

  • Generally speaking, the reader does not know what samples and how many samples were tested. It is also unknown what the result is (the average? From how many samples? What is the spread ?)

For XRD measurements, this is the average of 4 samples as described in chapter 2.5 and has also been added to the chapter 3.5. For calorimetric measurements, it is always a curve obtained during one measurement. For calorimetric measurements, only data obtained for one set of measurements at a time can be compared. Especially for isoperibolic calorimeters, different data can be obtained on different days because this method is strongly influenced by the ambient temperature. Despite all provisions such as the use of air conditioning this temperature may differ in summer and winter, but this method is very suitable for comparing samples.

  • Please provide the full uncertainty budget: A type, B type, standard and extended uncertainty.

As I mentioned above, the manuscript uses calorimetric data for one set of measurements. Only the general deviation of this measurement can be reported for this data. For calorimetric measurement using an isoperibolic calorimeter, the measurement deviation is around 10%. For isothermal calorimetry, the measurement is not so much affected and deviations are usually up to 5%.

Based on this comment, the sentence for XRD measurement was inserted into the work: The results show the averages of 4 measurements. It must be taken into account that this analysis is not completely accurate and errors can reach up to 5%.

The tables for XRD measurements are already very comprehensive. Adding additional values would make these tables confusing. I can also give the following tables which indicate the maximum variance of values from the average measured values:

The following Tables 7 - 10 shows the maximum variance of the measured values for 4 measurements from the value given in Tables 3 - 6.

Table 7. Deviation from the average for mineral composition of reference sample with cement and FA.

Hydration [day]

Ca3SiO5

Hatrurite (alite)

Ca2(SiO4)

Larnite (belite)

Ca2(Fe2O5)

Brownmillerite

C6A3H32

Ettringite

Ca(OH)2

Portlandite

Ca(CO3)

Calcite

(Mg0.67Al0.33(OH)2)(CO3)0.165
(H2O)0.48

Hydrotalcite

SiO2

Fe2O3

1

1.7

0.5

0.5

2.3

1.0

0.5

-

0.5

0.5

7

1.8

0.4

0.5

1.9

1.3

0.5

0.5

0.5

-

28

2.1

0.5

0.5

2.3

1.0

0.4

0.5

0.5

-

90

1.6

0.4

0.4

2.1

1.3

0.6

0.7

0.5

-

Table 8. Deviation from the average for mineral composition of sample with Zn(NO3)2∙6 H2O.

Hydration [day]

Ca3SiO5

Hatrurite (alite)

Ca2(SiO4)

Larnite (belite)

Ca2(Fe2O5)

Brownmillerite

C6A3H32

Ettringite

Ca[Zn2(OH)6](H2O)2

Ca(CO3)

Calcite

(Mg0.67Al0.33(OH)2)(CO3)0.165
(H2O)0,48

Hydrota.cite

3CaO∙Al2O3∙0,83Ca(NO3)2

0,17Ca(OH)2∙9,5H2O

SiO2

1

2.6

0.6

0.5

3.1

0.1

0.5

0.4

-

0.5

7

2.7

0.7

0.4

2.3

0.2

0.5

0.5

-

0.5

28

2.5

0.6

0.4

2.8

0.2

0.5

0.5

-

0.5

90

2.6

0.5

0.5

2.3

0.2

0.5

-

0.8

0.4

Table 9. Deviation from the average for mineral composition of sample with ZnCl2.

Hydration [day]

Ca3SiO5

Hatrurite (alite)

Ca2(SiO4)

Larnite (belite)

FeAlO3(CaO)2

Brownmillerite

C6A3H32

Ettringite

Ca2Al(OH)6Cl(H2O)2

Hydrocalumite

Zn5(OH)8Cl2H2O

Siminkolleite

SiO2

1

2.7

0.8

0.6

3.6

0.7

0.6

0.5

7

2.6

0.6

0.6

2.3

0.5

0.5

0.3

28

2.6

0.5

0.4

3.3

0.6

0.5

0.6

90

2.4

0.6

0.6

2.8

0.8

0.5

0.5

Table 10. Deviation from the average for mineral composition of sample with ZnO.

Hydration[day]

Ca3SiO5

Hatrurite (alite)

Ca2(SiO4)

Larnite (belite)

FeAlO3(CaO)2

Brownmillerite

C6A3H32

Ettringite

Ca(OH)2

Portlandite

Ca[Zn2(OH)6](H2O)2

ZnO

CaCO3

Calcite

CaSO4∙2H2O

Gypsum

SiO2

Ca(SO4)(H2O)0.5

Bassanite

Ca4Fe2O6CO3·12 H2O

(Mg0.67Al0.33(OH)2)(CO3)0.165
(H2O)0.48

Hydrotalcite

1

2.8

0.7

0.8

3.3

-

0.5

0.5

0.6

0.4

0.5

-

-

-

7

2.8

0.4

0.6

3.0

-

0.5

0.5

0.7

-

0.3

0.5

-

-

28

2.5

0.6

0.8

3.3

-

0.6

-

0.4

-

0.4

-

0.6

-

90

2.6

0.6

0.4

2.9

1.2

0.4

-

0.5

-

0.5

-

-

0.6

  • The equations 4 and 5 are presented in Figures 7. The verification of the statistical significance of the equations parameters is missing. Without this, it is not known whether the obtained parameters are the result of a coincidence or not.

The minimum accuracy expressed as R2 was 0.991 for isoperibolic measurements and 0.989 for isothermal measurements. These values are not added to the chart because they make this chart quite cluttered. A very good agreement of the results can be seen from the graph when the measured points correspond to the regression equation, which is shown in the text below the figure.

  • The equations in Figure 7 were obtained for 3 or 4 points. Is it sufficient to draw conclusions?

The resulting equations are usually created from four points. The results from measurement for the sample containing 1 % of Zn in the form of ZnCl2 was not included into the evaluation, since the acceleration of hydration, disalowing this way of evaluation, occured in large extent. These sentences are now mentioned in the text.

Thank you for this comment. Thanks to this comment, I found out that one of the graphs was inserted incorrectly. Now it's fixed.

  • Can the equations in Figure 7 be generalized in some way for future research (for research by other researchers) or are they just equations fitted to these historical results. If so, what for?

These equations show the mathematical quantification of a particular cement, fly ash and selected zinc compounds. If we use slightly different materials, the results may differ, but these equations are useful for predicting the behavior of similar materials.

Reviewer 3 Report

This work presents experimental results on the impact of zinc on the hydration of cement and the formation of mineral phases during cement hydration. The goal was to find out if or why some waste materials containing zinc are or might be detrimental to use in cement-based materials. The topic is of interest and timely. The manuscript presents useful data, but needs to be revised to improve its quality. Below are specific comments.

  1. General comment: grammar and punctuation needs to be improved.
    Lines 56-59: I am confused if the "poison" effect is beneficial or detrimental.
  2. Lines 75 to 86: poorly written compared to previous paragraphs.
  3. Table 1: the FA used has no zinc. Why this fly ash was used, rather than one that contained zinc, as the goal of the study is to see if zinc in fly ash causes problems for cement hydration and strength gain?
  4. Lines 174 to 225: this seems to be the "discussion", but it is coming before the actual results are presented. Either this should be better blended with the results, or placed as a separate discussion section after the results.
  5. Figures 1 to 7: the axis ranges (x- and y-axes) used in comparable figures are different, making comparison of graphs more difficult and potentially misleading. It is best to use the same axis ranges in comparable figures.
  6. Figure 7: commas should be decimal points.
  7. Section 3.3: should be 3.2, and likewise all subsequent sections need to move up by one.
  8. Sections 3.2 to 3.4: the text in these section can be made more concise.
  9. Lines 467 to 469: this comparison should be made graphically, comparing the present data with published data. Simply stating this and providing a reference is not informative for the reader.
  10. Lines 478 to 484: this is very little explanation of the data in Tables 3 to 6. While the previous sections were not concise enough, this is too concise. Just saying that one sample has more of some minerals, and another sample has more of other minerals, is not very informative.
  11. Conclusions: simply concluding that one type of zinc delays hydration while another accelerated hydration is not very informative for practical applications. The goal of the study was to find out if waste materials cause issues for cement hydration and mineralogy because of the presence of zinc. So what does the current study show? What waste materials are safe to use, and which are not. That is, what waste materials have the right type of zinc compound that causes no issues, while which waste materials are problematic?
  12. Conclusions: strength gain was not measured here. What do the current results say about the effect of different types of zinc in waste materials on strength gain? Some thoughts on this should be part of the conclusion.

Author Response

Reviewer 3

This work presents experimental results on the impact of zinc on the hydration of cement and the formation of mineral phases during cement hydration. The goal was to find out if or why some waste materials containing zinc are or might be detrimental to use in cement-based materials. The topic is of interest and timely. The manuscript presents useful data, but needs to be revised to improve its quality. Below are specific comments.

  1. General comment: grammar and punctuation needs to be improved.
    Lines 56-59: I am confused if the "poison" effect is beneficial or detrimental.

The text states that the poison effect causes significant hydration retardation, which is generally considered to be an negative effect. Lines 56-59 mainly explain how this retardation occurs. Based on this comment, the sentence “which is generally considered as a negative effect” has been added to the manuscript. Thank you for this comment.

  1. Lines 75 to 86: poorly written compared to previous paragraphs.

This section is written very simply due to the limitations of software-controlled plagiarism

  1. Table 1: the FA used has no zinc. Why this fly ash was used, rather than one that contained zinc, as the goal of the study is to see if zinc in fly ash causes problems for cement hydration and strength gain?

Fly ash, which was available at our workplace, was used for this research. The original effort of the whole project was mainly to use cement and fly ash without zinc and clearly show what happens when zinc is added in various forms. If the fly ash contained zinc, it would be a much more complex system that cannot be described without knowing how a clean system behaves. This is a very interesting comment that we will address in future research.

  1. Lines 174 to 225: this seems to be the "discussion", but it is coming before the actual results are presented. Either this should be better blended with the results, or placed as a separate discussion section after the results.

Yes, I fully agree that there is a discussion on these lines regarding the general behavior of all samples. For this reason, they are listed at the beginning of the Results and discussion section. This article was created as part of a project where articles already listed as literature 18 and 19 have already been published. If the opponent insists on this change, then of course this data can be blended with the results.

  1. Figures 1 to 7: the axis ranges (x- and y-axes) used in comparable figures are different, making comparison of graphs more difficult and potentially misleading. It is best to use the same axis ranges in comparable figures.

Based on this comment, all data for soluble compounds are reported for 140 hours. In the case of poorly soluble ZnO, data are given for 170 hours due to a greater effect on hydration. For soluble compounds, I cannot extend the data to 170 hours, as these compounds were measured for a shorter time than ZnO.

  1. Figure 7: commas should be decimal points.

Thank you for this comment, but I must apologize. I tried to find out how to change the decimal point to a dot in the graph axis. I did not find any instructions or option for this change in the Origin program. Even though it is an American company, I probably have only European version available, which probably does not allow this change

  1. Section 3.3: should be 3.2, and likewise all subsequent sections need to move up by one.

Thanks for reminder. Marking has been fixed.

  1. Sections 3.2 to 3.4: the text in these section can be made more concise.

Thanks for reminder. I will take this into account when writing the next manuscript. In this manuscript, I would like to leave the text a bit more extensive

  1. Lines 467 to 469: this comparison should be made graphically, comparing the present data with published data. Simply stating this and providing a reference is not informative for the reader.

Lines 467 to 469 contain the following information: "The results from isoperibolic measurement for the sample containing 1% of Zn in the form of ZnCl2 were not included into the evaluation, since the acceleration of hydration, disalowing this way of evaluation, occured in large extent. ”The opponent probably thinks a comparison with the published results on lines 473 - 475 below. I did not make this comparison due to problems with plagiarism of the results. However, if the reviewr insists on this comparison and the editors give me permission to do so because there is no problem with the presentation of already published results, then I will be happy to add this data in the form of a new graph where the data for fly ash and slag will be.

  1. Lines 478 to 484: this is very little explanation of the data in Tables 3 to 6. While the previous sections were not concise enough, this is too concise. Just saying that one sample has more of some minerals, and another sample has more of other minerals, is not very informative.

XRD identification was mainly used to explain why some changes in calorimetric curves occur due to the addition of zinc. This article is mainly focused on the use of isothermal and isoperibolic calorimetry to study the effect of zinc on hydration of cement blended with fly ash. This section is there only in addition to the results already mentioned in the discussions above. Based on this comment, the discussion was supplemented.

  1. Conclusions: simply concluding that one type of zinc delays hydration while another accelerated hydration is not very informative for practical applications. The goal of the study was to find out if waste materials cause issues for cement hydration and mineralogy because of the presence of zinc. So what does the current study show? What waste materials are safe to use, and which are not. That is, what waste materials have the right type of zinc compound that causes no issues, while which waste materials are problematic?

Some studies are published on both zinc added and zinc burned together with clinker. Both forms are dangerous. This study seeks to quantify in some way how much retardation occurs. Other research will also study materials that already contain zinc.

Based on this comment, the conclusion was modified

  1. Conclusions: strength gain was not measured here. What do the current results say about the effect of different types of zinc in waste materials on strength gain? Some thoughts on this should be part of the conclusion.

These considerations are now included in the conclusion

Reviewer 4 Report

In this paper, the author studied the effects of zinc in three different chemical compounds, which was ZnO, Zn(NO3)2 and ZnCl2 with fly ash on the hydration of Portland cement mainly using isothermal and isoperibolic calorimetry. For different combinations of cement and zinc compounds, the temperature and heat as a function of time in isoperibolic calorimetry as well as the heat flow and heat as a function of time in isothermal calorimetry are presented, indicating the mechanism of this phenomena. What’s more, the author also identified newly formed compounds by means of XRD and explained length of induction period. It is meaningful for further utilization of cement contaminated by zinc in various applications and helping optimize the amount of used secondary raw materials due to the fact that it illustated the effect of zinc components on the hydration of Portland cement.

However, the fatal weakness of this paper is the ambiguity of subject. I strongly recommend that this paper needs some clarifications or revisions before it can be accepted for publication in Materials.

  • Title: since this paper mainly discusses the effect of Zinc on hydration of cement blended with fly ash using isothermal and isoperibolic calorimetry and does not talk about the advantages and disadvantages of isothermal and isoperibolic calorimetry, I think it is better not to use “possibilities”, instead, directly use “with isothermal and isoperibolic calorimetry”.
  • Abstract: it is not suitable to introduce so many backgrounds in abstract and I think it is more preferable to put ‘Increasing...hydration process in cement’ into introduction and illustrate more about what you have done, what methods have you used and what the consequences you have got in your research.
  • Key words: there is no ‘isothermal calorimetry’ and ‘isoperibolic calorimetry’ in abstract, so it is not appropriate to use these words as your key words.
  • Introduction: in line 30, I think it is better to use ‘silicate’ instead of ‘silicon’.
  • Introduction:in line 77-78, you need to explain to the audience why there is a relationship with the amount of gypsum and C3
  • Materials and Methods: in line 119-120, although it is useful to investigate the effect of FA on the hydration of cement doped with zinc, I think using different kinds of zinc compounds can influence the outcome because as you say that the chlorine ions can accelerate the hydration and ZnO can decelerate the hydration due to its poor solubility.
  • Results and discussion: in line185-189, you indicated that the increasing first peak is the result of either zinc or high content of amorphous phase in fly ash. But if the reason is the later one, I think the peak will still increase without rising concentration or zinc.
  • In line 203-204, It is feasible to conclude that the decrease of heat flow due to higher amount of Zn2+ in amorphous phase. However, you did not explain why Zn2+ in amorphous phase can decrease of heat flow,which is important for revealing the mechanism of the interaction between Zn2+ and cement.
  • The language should be thoroughly checked. The manuscript has some spell and grammar errors. For example, in line 24, the ‘a’ in front of ‘1%’ should be ‘and’ ; in line 24, the last sentence in abstract should be past tense; in line 63, ‘cationts’ should be ‘cations’; in line 83, ‘they’ should be ‘the’; in line 88, ‘be marked as’ should be ‘be classified as’ ; in line 176, I do not know what do you want to express with ‘is lower also lower’.
  • There are numerical format errors in your manuscript. For instances, in line 22, the numbers in ‘ZnCl2 and Zn(NO)3’ do not use subscripts; in line 32, there are two blanks ahead of ‘tricalcium silicate’; in line 36, the format of S in 'C6AS3H32’ is wrong; in line 110, the ‘from’ should not use the black.

Author Response

Reviewer 4

In this paper, the author studied the effects of zinc in three different chemical compounds, which was ZnO, Zn(NO3)2 and ZnCl2 with fly ash on the hydration of Portland cement mainly using isothermal and isoperibolic calorimetry. For different combinations of cement and zinc compounds, the temperature and heat as a function of time in isoperibolic calorimetry as well as the heat flow and heat as a function of time in isothermal calorimetry are presented, indicating the mechanism of this phenomena. What’s more, the author also identified newly formed compounds by means of XRD and explained length of induction period. It is meaningful for further utilization of cement contaminated by zinc in various applications and helping optimize the amount of used secondary raw materials due to the fact that it illustated the effect of zinc components on the hydration of Portland cement.

However, the fatal weakness of this paper is the ambiguity of subject. I strongly recommend that this paper needs some clarifications or revisions before it can be accepted for publication in Materials.

  • Title: since this paper mainly discusses the effect of Zinc on hydration of cement blended with fly ash using isothermal and isoperibolic calorimetry and does not talk about the advantages and disadvantages of isothermal and isoperibolic calorimetry, I think it is better not to use “possibilities”, instead, directly use “with isothermal and isoperibolic calorimetry”.

Thank you for this comment, the title was changed as Use of Isothermal and Isoperibolic Calorimetry to Study the Effect of Zinc on Hydration of Cement Blended with Fly Ash

  • Abstract: it is not suitable to introduce so many backgrounds in abstract and I think it is more preferable to put ‘Increasing...hydration process in cement’ into introduction and illustrate more about what you have done, what methods have you used and what the consequences you have got in your research.

Thank you for the price comment, the abstract has been changed

  • Key words: there is no ‘isothermal calorimetry’ and ‘isoperibolic calorimetry’ in abstract, so it is not appropriate to use these words as your key words.

This article focuses on the use of isothermal and isoperibolic calorimetry. Therefore, they are used as key words. Abstract was rewritten based on comments. The words isothermal and isoperibolic calorimetry were used in the abstract. Thank you for the this comment.

  • Introduction: in line 30, I think it is better to use ‘silicate’ instead of ‘silicon’.

This designation is used as the official name of the element Si

  • Introduction:in line 77-78, you need to explain to the audience why there is a relationship with the amount of gypsum and C3

The following was added in the manuscript: One of given explanations is non-optimized gypsum content, which was based on the content and reactivity of C3A phase because overdosage in gypsum could induce a delay in cement hydration [16].

  • Materials and Methods: in line 119-120, although it is useful to investigate the effect of FA on the hydration of cement doped with zinc, I think using different kinds of zinc compounds can influence the outcome because as you say that the chlorine ions can accelerate the hydration and ZnO can decelerate the hydration due to its poor solubility.

This article focuses on the effect of selected compounds and the aim is to show the effect of these compounds. It is part of research focused on the effect of selected compounds.

  • Results and discussion: in line185-189, you indicated that the increasing first peak is the result of either zinc or high content of amorphous phase in fly ash. But if the reason is the later one, I think the peak will still increase without rising concentration or zinc.

Zinc reactions in the cement environment are a very complex mechanism. Zinc is able to change the control steps of reactions. Due to the formation of the zinc membrane, faster reactions of the amorphous phase can occur. This explanation was therefore supplemented by the sentence: This increase can be caused either by the effect of zinc, as a smaller amount of ettringite is formed compared to the hydration of OPC (ordinary portland cement), or by high content of amorphous phase in fly ash ( 78%), which can react faster thanks to another reaction mechanism due to the presence of zinc eg by the pozzolanic reaction [22, 29-31].

  • In line 203-204, It is feasible to conclude that the decrease of heat flow due to higher amount of Zn2+ in amorphous phase. However, you did not explain why Zn2+ in amorphous phase can decrease of heat flow,which is important for revealing the mechanism of the interaction between Zn2+ and cement.

Thanks for reminder. This explanation was supplemented as: The decrease of heat flow can be explained by higher amount of Zn2+ ions in amorphous phase because the formation of a zinc membrane (amorphous phase) reduces the reaction rates of some reactions, thereby reducing the heat flow.

  • The language should be thoroughly checked. The manuscript has some spell and grammar errors. For example, in line 24, the ‘a’ in front of ‘1%’ should be ‘and’ ; in line 24, the last sentence in abstract should be past tense; in line 63, ‘cationts’ should be ‘cations’; in line 83, ‘they’ should be ‘the’; in line 88, ‘be marked as’ should be ‘be classified as’ ; in line 176, I do not know what do you want to express with ‘is lower also lower’.

Thank you very much for your comments. These mistakes have been fixed.

  • There are numerical format errors in your manuscript. For instances, in line 22, the numbers in ‘ZnCl2 and Zn(NO)3’ do not use subscripts; in line 32, there are two blanks ahead of ‘tricalcium silicate’; in line 36, the format of S in 'C6AS3H32’ is wrong; in line 110, the ‘from’ should not use the black.

Thank you very much for your comments. These mistakes have been fixed.

Round 2

Reviewer 1 Report

The revisions are satisfactory.

Author Response

Thanks to the reviewer for your valuable comments

Reviewer 2 Report

I still believe that there is no statistical significance.
The verification of the statistical significance of the equations parameters is missing. Without this, it is not known whether the obtained parameters are the result of a coincidence or not.

It doesn't matter what the value of R or R ^ 2 is.
Statistical significance is important.
Suffice it to write that the statistical significance was checked.

The authors added and removed citations again
• Materials (2019), 12 (12), 1942, https://doi.org/10.3390/ma12121942 (this article is about fly ash in general, its various sources, etc.)
• Materials Science and Engineering: C (2019), Vol. 105, 110065, https://doi.org/10.1016/j.msec.2019.110065 (the article concerns the subject of the assessed work)
• Cement and Concrete Research (2015), vol. 72, pp. 128-136, https://doi.org/10.1016/j.cemconres.2015.02.023

Please fix the error.

Author Response

I still believe that there is no statistical significance.
The verification of the statistical significance of the equations parameters is missing. Without this, it is not known whether the obtained parameters are the result of a coincidence or not.

It doesn't matter what the value of R or R ^ 2 is.
Statistical significance is important.
Suffice it to write that the statistical significance was checked.

Thanks for the comment, the phrase "Statistical significance was checked for these equations" has been added.

The authors added and removed citations again
• Materials (2019), 12 (12), 1942, https://doi.org/10.3390/ma12121942 (this article is about fly ash in general, its various sources, etc.)

This citation was added as number 21

  • Materials Science and Engineering: C (2019), Vol. 105, 110065, https://doi.org/10.1016/j.msec.2019.110065 (the article concerns the subject of the assessed work)

This citation was added as number 14
• Cement and Concrete Research (2015), vol. 72, pp. 128-136, https://doi.org/10.1016/j.cemconres.2015.02.023

This citation was added as number 8

Thanks to the reviewer for your valuable comments

Reviewer 3 Report

Authors have addressed my concerns and made relevant revisions.

Author Response

(The authors gave the same response as above.)
